# Insights into catalysis and regulation of non-canonical ubiquitination and deubiquitination by bacterial deamidase effectors

Yong Wang[1,2,4], Qi Zhan[1,2,3,4], Xinlu Wang[1], Peipei Li[1,2,3], Songqing Liu[1,2], Guangxia Gao[1] & Pu Gao [1,2 ✉]

The bacterial effector MavC catalyzes non-canonical ubiquitination of host E2 enzyme UBE2N without engaging any of the conventional ubiquitination machinery, thereby abolishing UBE2N's function in forming K63-linked ubiquitin (Ub) chains and dampening NF-κB signaling. We now report the structures of MavC in complex with conjugated UBE2N~Ub and an inhibitor protein Lpg2149, as well as the structure of its ortholog, MvcA, bound to Lpg2149. Recognition of UBE2N and Ub depends on several unique features of MavC, which explains the inability of MvcA to catalyze ubiquitination. Unexpectedly, MavC and MvcA also possess deubiquitinase activity against MavC-mediated ubiquitination, highlighting MavC as a unique enzyme possessing deamidation, ubiquitination, and deubiquitination activities. Further, Lpg2149 directly binds and inhibits both MavC and MvcA by disrupting the interactions between enzymes and Ub. These results provide detailed insights into catalysis and regulation of MavC-type enzymes and the molecular mechanisms of this non-canonical ubiquitination machinery.

[1] CAS Key Laboratory of Infection and Immunity, CAS Center for Excellence in Biomacromolecules, Institute of Biophysics, Chinese Academy of Sciences, Beijing 100101, China. [2] National Laboratory of Biomacromolecules, Institute of Biophysics, Chinese Academy of Sciences, Beijing 100101, China. [3] University of Chinese Academy of Sciences, Beijing 100049, China. [4] These authors contributed equally: Yong Wang, Qi Zhan. ✉email: gaopu@ibp.ac.cn

Ubiquitination is an important post-translational modification that regulates most if not all cellular processes[1,2]. Conventional ubiquitination is carried out by the E1–E2–E3 three-enzyme cascade in an ATP-dependent manner, resulting in a covalent link formed between the carboxy terminus of ubiquitin (Ub) and an acceptor lysine residue on substrates[3]. Subsequent Ub moieties can be added to the seven lysine residues or the N-terminal methionine on the proximal Ub to generate functionally diverse poly-Ub chains[2]. Ubiquitination can be removed by specific enzymes called deubiquitinases (DUBs), thus maintaining the dynamic state of the cellular ubiquitome[4].

Given the broad importance of Ub signaling, it is not surprising that a variety of microbial pathogens adopt intricate mechanisms to interfere with the host ubiquitination network[5–7]. Such interference by pathogen-derived virulence factors is normally dependent on the host conventional ubiquitination system. Strikingly, the SidE- and MavC-type enzymes, two recently identified effector families from the bacterial pathogen Legionella pneumophila, can ubiquitinate host substrates through non-canonical mechanisms without engaging any of the conventional ubiquitination machinery[8–12]. Ubiquitination catalyzed by SidE-type enzymes requires the cofactor NAD and relies on the concerted action of two enzymatic domains: the mono-ADP-ribosyltransferase (mART) domain that transfers ADP-ribose (ADPr) moiety of NAD to Ub R42, and the phosphodiesterase (PDE) domain that cleaves ADPr-Ub to AMP and phospho-ribosylated Ub (Pr-Ub), conjugating Pr-Ub to substrate serine[13–16]. Later studies further revealed the regulation mechanisms of SidE-mediated ubiquitination by two kinds of L. pneumophila effectors, SidJ and DupA/DupB, which act as either a glutamylase to inhibit SidE enzymatic activity[17–20] or DUBs to remove the Pr-linked ubiquitination[21,22], respectively.

MavC was initially found to be a homolog of the Ub/NEDD8-deamidase Cif (cycle inhibiting factor)[12,23]. In contrast to Cif family enzymes that can deamidate a conserved glutamine residue (Q40 to E40) primarily on NEDD8 and in certain cases on Ub, MavC specifically deamidates Q40 of Ub[12]. More recently, MavC was also shown to specifically interact with the host E2 protein UBE2N, and catalyze monoubiquitination of UBE2N by forming a covalent linkage between Q40 of Ub and K92 of UBE2N[9,12]. This non-canonical ubiquitination activity of MavC has never been reported for Cif and other deamidase families, thus representing the first example of an enzyme that couples both deamidation and ubiquitination activities. UBE2N functions as the major E2 enzyme in host cells that generates canonical K63-linked Ub chains, which are critical to signaling in inflammatory (e.g., NF-κB) and DNA damage response pathways[24–26]. The MavC-mediated non-canonical ubiquitination of UBE2N has been shown to abolish its E2 activity and dampen the NF-κB pathway in the early phase of L. pneumophila infection[9,12]. The genomic cluster of MavC encodes three effector proteins: Lpg2147 (MavC itself), Lpg2148 (MavC paralog A, or MvcA), and Lpg2149[12]. Although MvcA shares a high degree of similarity with MavC and possesses Ub-deamidase activity, it cannot catalyze UBE2N ubiquitination[9]. Interestingly, Lpg2149, a small 119-residue protein, was shown to interact with MavC and MvcA and inhibit the deamidase activities of both proteins[12].

In contrast to SidE-type enzymes that have been extensively studied, many critical questions related to the MavC system remain to be explored, including the molecular basis of substrate (UBE2N) and Ub recognition by MavC, the mechanisms of Lpg2149-mediated inhibition for MavC and MvcA, the rationale of different activities for MavC and MvcA, the structural basis of inhibition for UBE2N E2 activity by MavC-mediated ubiquitination, as well as the identification and characterization of potential DUBs that can remove MavC-mediated ubiquitination. In this work, we conduct systematic biochemical and structural analysis and determine the mechanisms of the above processes. Unexpectedly, our data reveal that both MavC and MvcA also function as DUBs to remove the MavC-mediated ubiquitination, thus highlighting a unique enzyme family possessing deamidation, ubiquitination, and deubiquitination activities.

## Results

**Structure of MavC bound to conjugated UBE2N~Ub**. To understand the molecular mechanism of MavC-mediated ubiquitination of UBE2N, we have solved the 2.7 Å crystal structure of MavC$^{\Delta CTD}$ carrying an inactivating mutation (C74A) in complex with a pre-conjugated UBE2N~Ub, which is prepared by treating separated Ub and UBE2N with the wild-type (WT) MavC enzyme (Fig. 1a–c and Supplementary Fig. 1a–c; X-ray statistics in Table 1). There is one MavC and one UBE2N~Ub in the asymmetric unit, highlighting a 1:1:1 molar ratio of the ternary complex (Fig. 1b, c). MavC resembles an overall crab claw architecture, with the Insertion and Catalytic domains forming one half, and the helix-bundle domain (HBD) forming the other (Fig. 1b). Ub inserts deeply into the crab claw cleft and is tightly sandwiched between the Catalytic and HBD domains (Fig. 1b, c). UBE2N attaches to one side of MavC and form interactions with both Insertion and Catalytic domains (Fig. 1b, c). Although the relative orientation between Insertion and Catalytic domains can be dynamic due to the intrinsically flexible linkers connecting them, the binding to UBE2N may force these two domains to adopt a fixed inter-domain conformation. Indeed, structural superimposition of MavC in apo[12] and UBE2N~Ub bound states shows that the Insertion domain undergoes a ~30° rotation upon binding to UBE2N (Fig. 1d).

Given that the pre-conjugated UBE2N~Ub was used for crystallization, the current structure represents a product-bound form of MavC. The electron density clearly shows that an

| | MavC +UBE2N-Ub | MavC +Lpg2149 | MvcA +Lpg2149 |
|---|---|---|---|
| **Table 1 Data collection and refinement statistics.** | | | |
| **Data collection** | | | |
| Space group | $P6_5$ | $P2_12_12_1$ | $C2$ |
| Cell dimensions | | | |
| $a, b, c$ (Å) | 150.5, 150.5, 58.4 | 58.7, 68.8, 138.4 | 137.0, 81.2, 54.3 |
| $\alpha, \beta, \gamma$ (°) | 90, 90, 120 | 90, 90, 90 | 90, 95.1, 90 |
| Resolution (Å) | 50-2.70 (2.75-2.70) | 50-2.70 (2.76-2.70) | 50-2.70 (2.76-2.70) |
| $R_{merge}$ | 0.112 (0.541) | 0.125 (0.524) | 0.064 (0.383) |
| $I/\sigma (I)$ | 29.0 (3.0) | 15.5 (3.2) | 21.6 (4.3) |
| Completeness (%) | 100.0 (100.0) | 99.8 (99.9) | 96.7 (97.5) |
| Redundancy | 14.9 (13.2) | 9.6 (9.6) | 5.3 (5.5) |
| **Refinement** | | | |
| Resolution (Å) | 25.1-2.7 | 37.2-2.7 | 24.8-2.7 |
| No. reflections | 36,277 | 14,489 | 15,703 |
| $R_{work}/R_{free}$ | 0.193/0.243 | 0.205/0.257 | 0.215/0.258 |
| No. atoms | | | |
| Protein | 4800 | 3859 | 3815 |
| Ligand/ion | 0 | 0 | 4 |
| Water | 75 | 49 | 62 |
| B-factors | | | |
| Protein | 42.8 | 42.8 | 50.5 |
| Ligand/ion | – | – | 44.2 |
| Water | 33.6 | 30.7 | 30.6 |
| RMS deviations | | | |
| Bond lengths (Å) | 0.003 | 0.004 | 0.006 |
| Bond angles (°) | 0.586 | 0.652 | 0.740 |

Values in parentheses are for the highest-resolution shell.

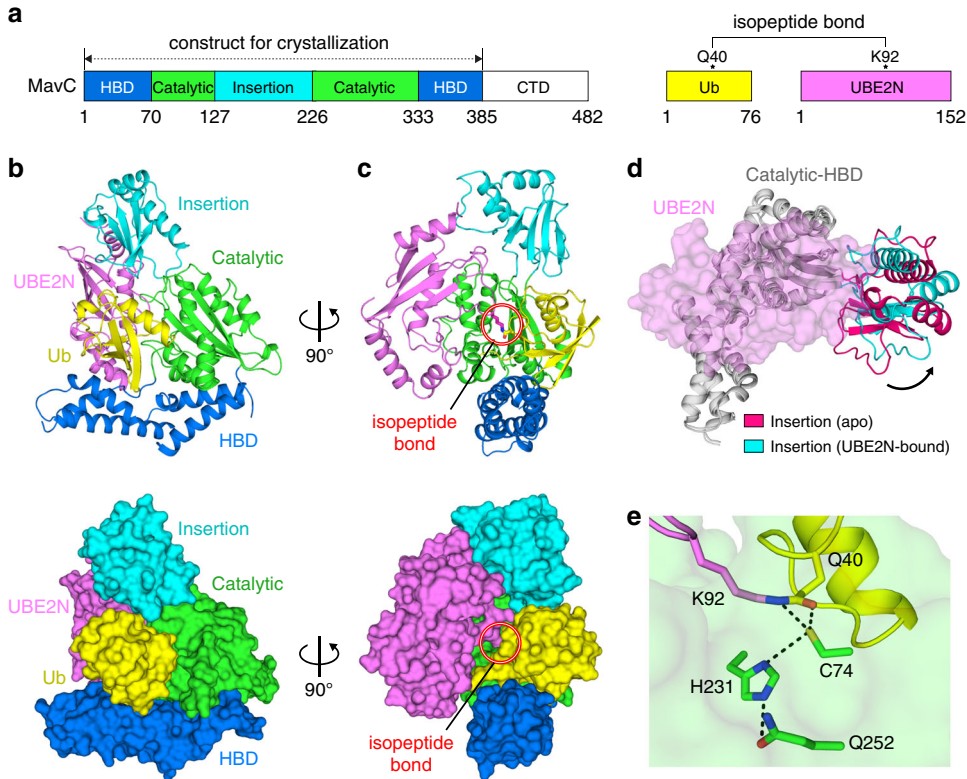

**Fig. 1 Structure of MavC in complex with conjugated UBE2N~Ub. a** Domain organization of MavC and conjugated UBE2N~Ub. **b, c** Two views of cartoon (upper) and surface (lower) representations of the MavC+UBE2N~Ub complex, with the same color code as in **a**. The isopeptide bond between UBE2N-K92 and Ub-Q40 was highlighted in red circles. **d** Structural superimposition of MavC in apo (PDB: 5TSC) and UBE2N-bound states and conformational changes of the Insertion domain upon binding to UBE2N. **e** Detailed hydrogen bonding interactions between MavC catalytic residues and the K92–Q40 isopeptide.

isopeptide bond is formed between the side-chains of UBE2N-K92 and Ub-Q40, and at the same time the amino group of Ub-Q40 has been removed (Fig. 1e and Supplementary Fig. 1d). These structural findings confirm the previous predictions of MavC-mediated ubiquitination on UBE2N[9]. Moreover, the deamidase active site of MavC is located right below the isopeptide bond and the catalytic residues, particularly C74 (modeled back from A74), can form extensive interactions with both K92 and Q40 (Fig. 1e). These findings confirm that MavC catalyzes both Ub deamidation and UBE2N ubiquitination by using the same pocket.

**Interactions between MavC and UBE2N.** UBE2N forms inter-molecular interactions with both the Catalytic (630 Å$^2$) and Insertion (563 Å$^2$) domains of MavC (Fig. 2a). The UBE2N–Catalytic interactions are mainly mediated by a loose loop region (aa 87–100) of UBE2N, which perfectly fits within a groove of MavC (Fig. 2b). Several important residues of MavC, with bulky side-chains, constitute the surface of this UBE2N-binding groove and form both hydrogen bonds and hydrophobic contacts with UBE2N (Fig. 2b and Supplementary Fig. 2a). Specifically, the side-chain of M317 from MavC inserts into a hydrophobic pocket composed of residues from both the aa 87–100 loop (L88, I90, W95, L99) and its connecting helix (V104) of UBE2N (Fig. 2b, c). Thus, M317 acts as a critical anchor point stabilizing the MavC–UBE2N interactions. It should be noted that the aa 87–100 region of UBE2N adopts a relatively compact conformation in previously reported structures (both apo and partner-bound states), which positions K92 within a small α-helix and hinders potential ubiquitination on its side-chain (Fig. 2d and Supplementary Fig. 2b). Through the

interactions with MavC, this region undergoes considerable changes, disrupting the helix structure and forming a more stretched conformation in which K92 is accessible to both the catalytic pocket of MavC and Q40 of Ub (Fig. 2d and Supplementary Fig. 2b).

The UBE2N–Insertion surface is formed between a large loop region (aa 185–207) of MavC and three small regions from one end of UBE2N (P5/R6/R7, P63/M64, and P97/Q100) (Fig. 2a, e and Supplementary Fig. 2c). This inter-molecular binding is mainly contributed by hydrophobic interactions (Fig. 2e) and further strengthened by a few hydrogen bonds (Supplementary Fig. 2c).

To evaluate the potential effects of above MavC–UBE2N interactions, we generated multiple MavC mutants and tested their ubiquitination activities. Alanine substitutions for most residues on the UBE2N–Catalytic surface (R126A, Y254A, K320A, Y300A, F313A, M317A, F321A) significantly decrease the ubiquitination activity (Fig. 2f). Similarly, mutations that disrupt the UBE2N–Insertion contacts (F188A/Y189A/Y192A, Y198A) also dramatically reduce the activity (Fig. 2g) and in the meantime abolish the binding (Supplementary Fig. 2d). In line with the MavC mutants, alanine substitutions for UBE2N on both interacting surfaces show notable reduction in ubiquitination efficiency (Supplementary Fig. 2e). Together, these results highlight the importance of MavC–UBE2N interactions for MavC-mediated UBE2N ubiquitination.

**Unconserved UBE2N–Insertion surface of MvcA.** Although MavC and its close ortholog MvcA share considerably high sequence identity (~52%) and both can catalyze Ub-Q40 deamidation, MvcA cannot ubiquitinate UBE2N[9,12]. We anticipated

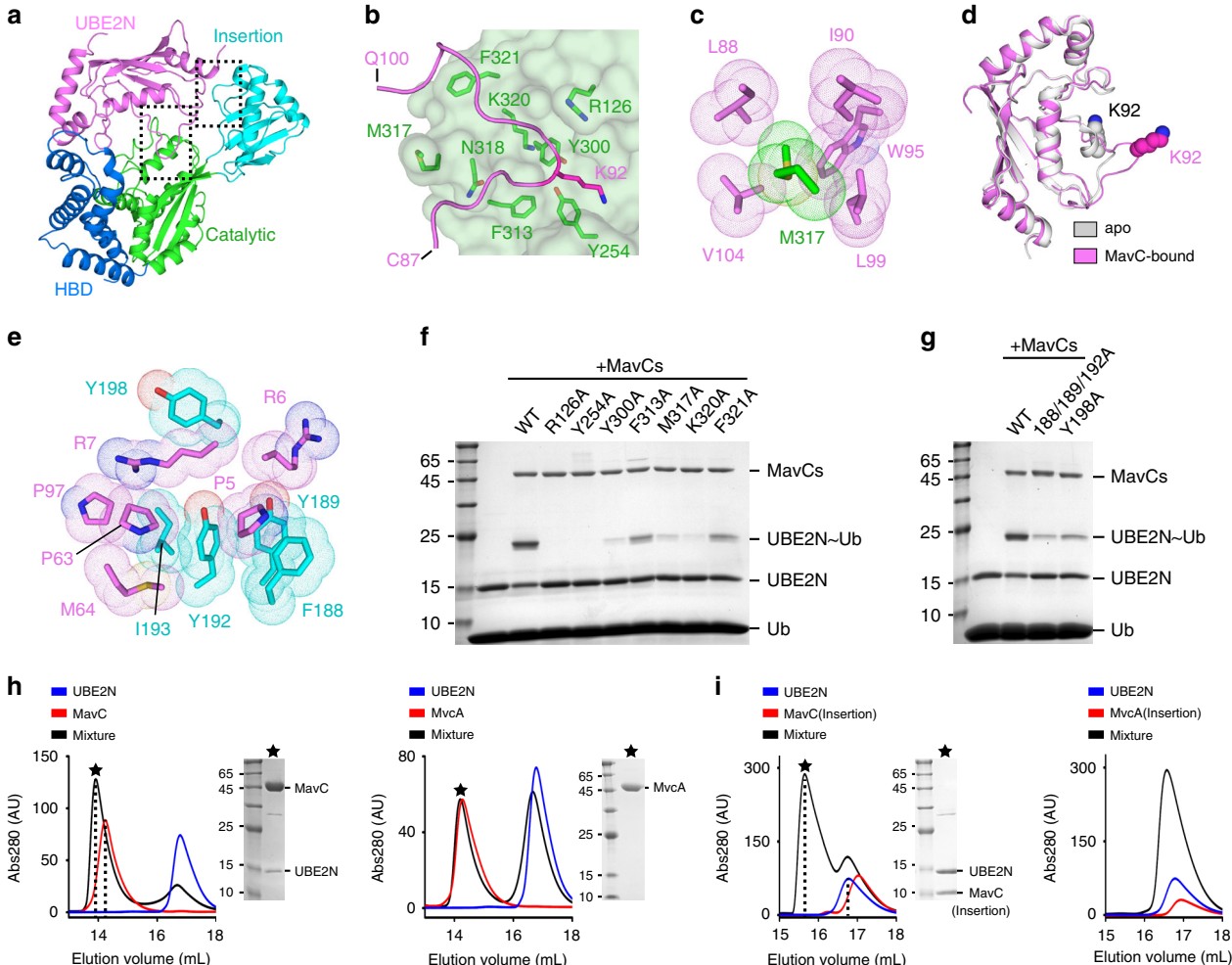

**Fig. 2 Interactions between MavC and UBE2N. a** Overall view of UBE2N–Catalytic and UBE2N–Insertion interacting surfaces, with the same color code as in Fig. 1a. **b** The aa 87–100 loop region of UBE2N (cartoon; purple) binds within a critical groove of MavC Catalytic domain (surface; green). **c** Detailed hydrophobic interactions between MavC-M317 (green) and the residues from UBE2N (purple). **d** Conformational changes of UBE2N upon binding to MavC. The K92 residues were highlighted as spheres. **e** Detailed hydrophobic interactions between MavC Insertion domain (cyan) and UBE2N (purple). **f**, **g** In vitro ubiquitination reactions for various MavC mutants in both Catalytic (**f**) and Insertion (**g**) domains. The reaction mixtures were subjected to SDS-PAGE and stained with Coomassie dye. **h**, **i** Elution profiles of SEC runs on Superdex 200 10/300 column to test binding of UBE2N with MavC/MvcA (**h**) or their isolated Insertion domains (**i**). Note that only MavC or MavC–Insertion, but not MvcA or MvcA–Insertion, has a considerable shift in elution volume upon binding to UBE2N. Black asterisks indicate the fractions analyzed by SDS-PAGE. Source data are provided as a Source Data file. Experiments in **f**–**i** were repeated independently three times with similar results.

that the UBE2N–MavC interactions found in the crystal structure may not be conserved for MvcA. Indeed, although the UBE2N–Catalytic contacts are largely maintained in MvcA, several key residues forming UBE2N–Insertion surface are not conserved in MvcA (Supplementary Fig. 2f). Size-exclusion chromatography (SEC) results show that while MavC can form a stable complex with UBE2N, MvcA cannot (Fig. 2h). Further SEC assays using the isolated Insertion domains instead of full-length proteins also show that only MavC Insertion domain can form a complex with UBE2N (Fig. 2i). These data provide a detailed explanation of the different enzymatic activities for MavC and MvcA.

**Interactions between MavC and Ub.** Ub binds tightly within the cleft between Catalytic and HBD domains (Fig. 3a) through both hydrogen bonding and hydrophobic interactions. The hydrogen bonding interactions are mainly involved in three regions: (1) a helix-loop motif of Ub (aa 31–40), where the Q40 resides, forms extensive contacts with both the catalytic

residues (Fig. 1e) and other residues surrounding the active pocket (Fig. 3b); (2) the central β-sheet region of Ub which forms two pairs of hydrogen bonds with N39 and E42 from HBD (Fig. 3c); and (3) the C-terminal tail of Ub (aa 71–74) which makes contacts with residues from the connecting region (aa 70–73) of Catalytic and HBD domains (Fig. 3d). In addition, there are two notable hydrophobic surfaces between Ub and MavC. The first surface is mainly mediated by W255 from MavC, which inserts into a compact pocket formed by both main- and side-chains of residues from both proteins (Fig. 3e). The second surface adopts an intersected arrangement by L8/H68 from Ub and L36/I43 from MavC (Fig. 3f). It should be noted that although the C-terminal tail of Ub forms both hydrogen bonding and hydrophobic interactions with MavC (Fig. 3d, e), the two glycines at Ub C-terminus that are critical for conventional ubiquitination are disordered in the structure and point into the solvent (Supplementary Fig. 3a). This, again, confirms that MavC-mediated ubiquitination is distinct from the conventional process.

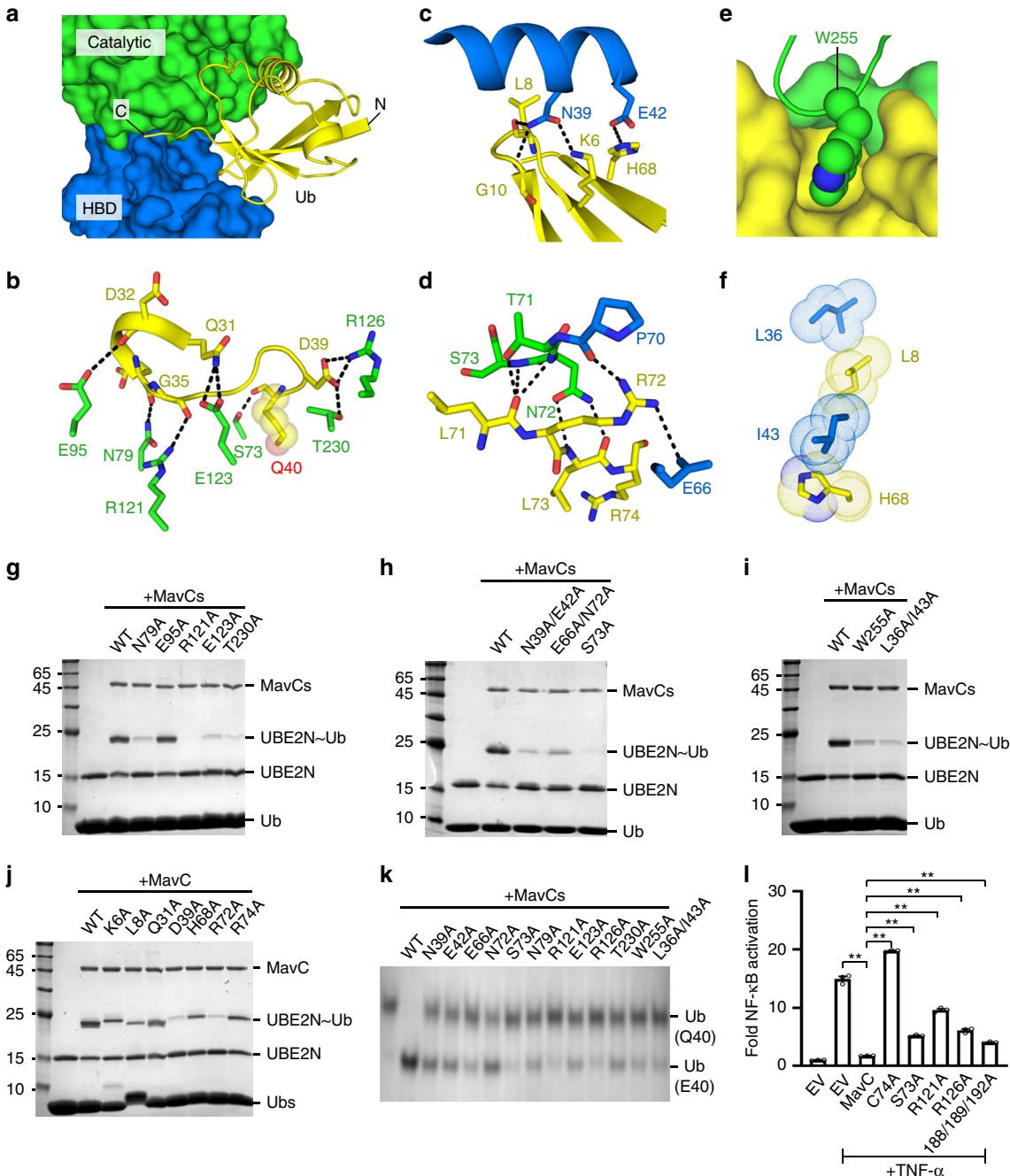

**Fig. 3 Interactions between MavC and Ub. a** Overall view of Ub (cartoon; yellow) binding within the cleft of Catalytic (surface; green) and HBD (surface; blue) domains of MavC. **b–d** Three major regions of hydrogen bonding interactions between MavC and Ub, same color code as in **a**. The hydrogen bonds were shown in black dash lines. **e**, **f** Two major regions of hydrophobic interactions between MavC and Ub, same color code as in **a**. **g–i** In vitro ubiquitination reactions for various MavC mutants on different MavC–Ub inter-molecular interacting surfaces. The reaction mixtures were subjected to SDS-PAGE and stained with Coomassie dye. **j** Indicated Ub proteins were incubated with MavC and UBE2N. The reaction mixtures were subjected to SDS-PAGE and stained with Coomassie dye. **k** In vitro Ub deamidation reactions for various MavC mutants. The reaction mixtures were subjected to native-PAGE and stained with Coomassie dye. **l** Mutations on MavC decreased its inhibition ability to NF-κB activation in cells. Plasmids encoding MavC or its mutants were co-transfected with NF-κB luciferase reporter system plasmids. Cells were mock-treated or treated with TNF-α. The luciferase activities were measured for the activation of NF-κB reporter (see details in Methods). Data presented are means ± SEM of three independent experiments. **\*\*$p < 0.01$, unpaired and two-tailed $t$ test. Source data are provided as a Source Data file. Experiments in **g–l** were repeated independently three times with similar results.

To validate the importance of MavC–Ub interactions, we generated multiple MavC mutants covering all of the above interacting surfaces and analyzed their ubiquitination activities. Alanine substitutions for most key residues (except E95A) of the first hydrogen bonding surface (Fig. 3b) dramatically inhibit

UBE2N ubiquitination (Fig. 3g). Similarly, mutations on the second (Fig. 3c) and third (Fig. 3d) hydrogen bonding surfaces notably reduce the ubiquitination activities (Fig. 3h). Furthermore, alanine substitutions of the two hydrophobic surfaces (W255A and L36A/I43A) also inhibit UBE2N ubiquitination

(Fig. 3i). In line with the MavC mutants, mutations of Ub on the inter-molecular surfaces show notable reduction in ubiquitination activity, albeit with different efficiency (Fig. 3j). Taken together, these data highlight the critical roles of MavC–Ub interactions on UBE2N ubiquitination.

**Ub-interacting surface is critical for deamidation**. Given that MavC employs the same active site to catalyze Ub deamidation and UBE2N ubiquitination (Fig. 1e), we anticipated that the MavC–Ub interactions should be important for both processes. Indeed, the above MavC and Ub mutants reduce Ub deamidation efficiency to a similar degree as the reduction for UBE2N ubiquitination (Fig. 3k and Supplementary Fig. 3b). To see whether MvcA shares a similar Ub-interacting surface with MavC, we generated mutations on MvcA at the equivalent positions as those for MavC mutants (Supplementary Fig. 3c). As expected, the mutations of both MvcA and Ub also inhibit MvcA-mediated Ub deamidation (Supplementary Fig. 3c, d), confirming that MvcA and MavC share a similar mechanism for Ub deamidation.

**Molecular interactions are important for NF-κB inhibition**. It has been shown that the ubiquitination of UBE2N catalyzed by MavC abolishes its activity in the formation of K63-linked Ub chains, which further dampens NF-κB signaling in the initial phase of bacterial infection[9]. To confirm the importance of the inter-molecular interactions found in our crystal structure in a cellular context, we used a luminescent reporter assay where HEK293T cells were co-transfected with an NF-κB luciferase reporter vector along with a vector expressing MavC (WT or mutants). In line with the previous studies[9,12], MavC, but not MavC[C74A], inhibits both the formation of K63-linked Ub chains (Supplementary Fig. 3e) and the TNF-α (tumor necrosis factor alpha) induced NF-κB activation (Fig. 3l). Moreover, the four MavC mutants (S73A, R121A, R126A, F188A/Y189A/Y192A) that partially disrupt the MavC–UBE2N and MavC–Ub interactions show significant reduction of the NF-κB inhibition (Fig. 3l). Consistent with the results of NF-κB luciferase reporter assays, the immunoblotting experiments confirm that MavC[C74A] completely abolishes the Ub ligase activity and the other four MavC mutants can partially ubiquitinate UBE2N (Supplementary Fig. 3f). These results highlight the critical roles of these inter-molecular interactions in both in vitro UBE2N ubiquitination and cellular NF-κB inhibition assays. Clearly, future works will be needed to validate the effects of these structure-based mutants in the context of *L. pneumophila* infection.

**UBE2N ubiquitination can be removed by MavC itself**. When performing the in vitro MavC-mediated ubiquitination assays, we consistently observed that the amount of newly synthesized UBE2N~Ub gradually decreased over time (Supplementary Fig. 4a). We initially anticipated that UBE2N~Ub may be unstable by itself, however, the purified UBE2N~Ub can stay unchanged over a long period of time (Supplementary Fig. 4b). Although MavC belongs to the Cif-type Ub-deamidases[12,23], its ability to catalyze the formation of an isopeptide between glutamine from one protein and lysine from another is reminiscent of the activity of transglutaminases[9,27]. Given that certain members of the transglutaminase family can catalyze both formation and cleavage of the isopeptide[27], we hypothesized that MavC may also act as a DUB to remove the UBE2N ubiquitination. Indeed, upon incubating with MavC, the purified UBE2N~Ub can be readily cleaved into two parts that have similar molecular weights to the separated UBE2N and Ub (Fig. 4a). In addition, MavC[C74A] mutant no longer has this activity (Fig. 4a), indicating that MavC uses the same active pocket to catalyze deubiquitination. We then

hypothesized that the isopeptide bond between Ub and UBE2N is the cleavage site. In this case, the cleavage products will be a WT UBE2N and a deamidated Ub (Q40E). As expected, further mass spectrometry experiment clearly shows that the Ub molecule after MavC-mediated cleavage is indeed the deamidated Ub (Q40E) (Fig. 4b).

Since MavC can ubiquitinate and deubiquitinate UBE2N, our crystal structure of MavC in complex with the conjugated UBE2N~Ub (Fig. 1b, c) can represent both the product-bound state of the ubiquitination process and the substrate-bound state of the deubiquitination process. Thus, the MavC–UBE2N and MavC–Ub interactions that show to be important for MavC-mediated ubiquitination (Figs. 2a–e and 3a–f) will also be critical for MavC-mediated deubiquitination. Indeed, the mutations of MavC on these inter-molecular surfaces also decrease the deubiquitination activity of MavC, albeit with different efficiency (Fig. 4c, d). Taken together, our data show that MavC can catalyze deamidation, ubiquitination, and deubiquitination by using the same active pocket.

**MvcA can also remove MavC-mediated UBE2N ubiquitination**. Although MvcA cannot ubiquitinate free UBE2N due to the unconserved UBE2N–Insertion surface, it still contains the Ub-binding and UBE2N–Catalytic interacting surfaces, which may enable MvcA to interact with the conjugated UBE2N~Ub and catalyze the deubiquitination reaction. Indeed, SEC experiments show that MvcA can form a stable complex with the conjugated UBE2N~Ub (Supplementary Fig. 4c), although it does not bind free UBE2N in the same condition (Fig. 2h). We then incubated both WT MvcA and an inactive mutant MvcA[C83A] (a corresponding mutant to MavC[C74A]) with UBE2N~Ub and found that WT MvcA does have robust DUB activity while MvcA[C83A] does not (Fig. 4e). Next, we generated several mutations for MvcA at the potential MvcA–UBE2N and MvcA–Ub surfaces based on MavC+UBE2N~Ub structure (Supplementary Fig. 2f), and tested their effects for deubiquitination. As expected, these mutants show considerable reduction of their DUB activities (Fig. 4f). Furthermore, the mass spectrometry experiment shows that the Ub molecule, after MvcA cleavage, is Ub (Q40E) (Fig. 4g), confirming that the UBE2N–Ub isopeptide bond is also the cleave site of MvcA. These data show that MvcA can deubiquitinate UBE2N~Ub using the same mechanism as MavC.

To compare the DUB activities between MavC and MvcA, we conducted the time-dose deubiquitination assays by using the same amounts of enzymes and substrates (UBE2N~Ub) (Supplementary Fig. 4d, e, and Fig. 4h). Unexpectedly, MvcA shows a much more robust DUB efficiency than MavC (Fig. 4h), with a 50% completion of cleavage within ~10 s, for which MavC needs ~10 min (Supplementary Fig. 4d, e). We anticipated that MvcA may have an intrinsically higher DUB activity than MavC, therefore, MvcA can catalyze the cleavage reaction more efficiently once the conjugated UBE2N~Ub binds to the enzyme sufficiently strongly. In addition, although the formation and cleavage of this non-canonical ubiquitination are catalyzed within the same pocket, the ligase and DUB activities are not necessarily coupled with each other. Such a dramatic difference in deubiquitination activities indicates that MvcA, rather than MavC, may act as the major DUB for MavC-mediated non-canonical ubiquitination. In line with this hypothesis, a very recent work focusing on MvcA also reports the DUB activity of MvcA and its ability to reverse MavC-induced UBE2N ubiquitination in the context of bacterial infections[28]. Clearly, future studies will be needed to fully understand why MvcA has much higher DUB activity than MavC.

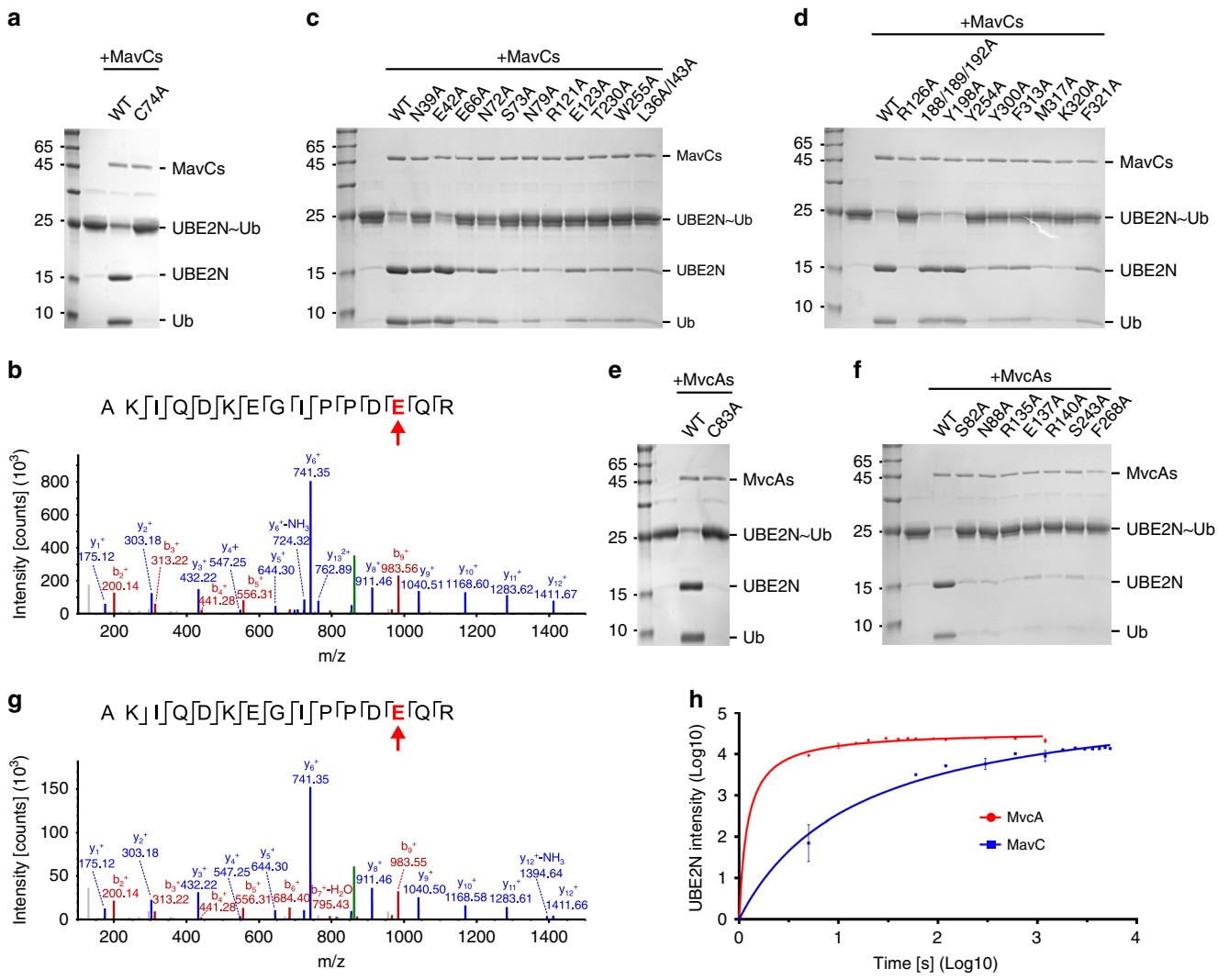

**Fig. 4 MavC and MvcA can remove MavC-mediated ubiquitination. a** The conjugated UBE2N~Ub can be cleaved by WT MavC, but not MavC[C74A]. **b** Mass spectrometry analysis of the protein band corresponding to Ub catalyzed by WT MavC from panel **a**. The Ub product is the deamidated form (E40). **c, d** Indicated MavC proteins were incubated with the conjugated UBE2N~Ub. The reaction mixtures were subjected to SDS-PAGE and stained with Coomassie dye. **e** The conjugated UBE2N~Ub can be cleaved by WT MvcA, but not MvcA[C83A]. **f** Indicated MvcA proteins were incubated with the conjugated UBE2N~Ub. The reaction mixtures were subjected to SDS-PAGE and stained with Coomassie dye. **g** Mass spectrometry analysis of the protein band corresponding to Ub catalyzed by WT MvcA from panel **e**. The Ub product is the deamidated form (E40). **h** The conjugated UBE2N~Ub was treated with same amounts of MavC and MvcA. The intensities of released UBE2N bands were measured in different time points (shown in Supplementary Fig. 4d, e). Each experiment was done in duplicate, and the data were analyzed using GraphPad Prism software. The error bars represent mean ± SEM. Source data are provided as a Source Data file. Experiments in **a**, **c**–**f** were repeated independently three times with similar results.

**Molecular basis of inhibition for MavC and MvcA by Lpg2149.** In addition to MavC and MvcA, the same genomic cluster in *L. pneumophila* also encodes a small effector protein, Lpg2149, which has been shown to interact with both enzymes and inhibit their deamidation activities[12]. The inhibitory role of Lpg2149 in the MavC system is reminiscent of the function of SidJ in the SidE system[17–20], although their operation mechanisms are distinct from each other. To understand the detailed molecular mechanisms of Lpg2149-mediated inhibition, we determined the crystal structures of MavC–Lpg2149 and MvcA–Lpg2149 complexes (Fig. 5a, b; X-ray statistics in Table 1). The asymmetric unit contains one Lpg2149 and one enzyme for both structures, highlighting a 1:1 molar ratio of the complex formation. Both structures adopt a similar assembly, with the Lpg2149 monomer binding to the connecting region between Catalytic and HBD domains (Fig. 5a, b). Lpg2149 forms similar and extensive hydrogen bonding (Fig. 5c and Supplementary Fig. 5a) and

hydrophobic (Fig. 5d and Supplementary Fig. 5b) interactions with MavC and MvcA, which are mainly mediated by the N-terminal α-helix (aa 11–37) and a helix-loop motif (aa 74–86) of Lpg2149. The similar interaction patterns of Lpg2149 with both enzymes reveal that Lpg2149 employs a conserved mechanism to inhibit MavC and MvcA.

Structural superimposition of MavC in Ub- and Lpg2149-bound states shows that Lpg2149 and Ub occupy a similar binding position and form severe steric clashes with each other (Fig. 5e), indicating that Lpg2149 can interfere with the interactions between Ub and MavC/MvcA. Indeed, SEC experiments show that Lpg2149 can replace the conjugated UBE2N~Ub from the preformed MavC[C74A]+UBE2N~Ub and MvcA[C83A]+UBE2N~Ub complexes, albeit with different efficiency (Supplementary Fig. 5c). Based on the structural information, Lpg2149 should have a general inhibition ability for not only Ub deamidation (by MavC/MvcA), but also UBE2N ubiquitination (by MavC) and deubiquitination

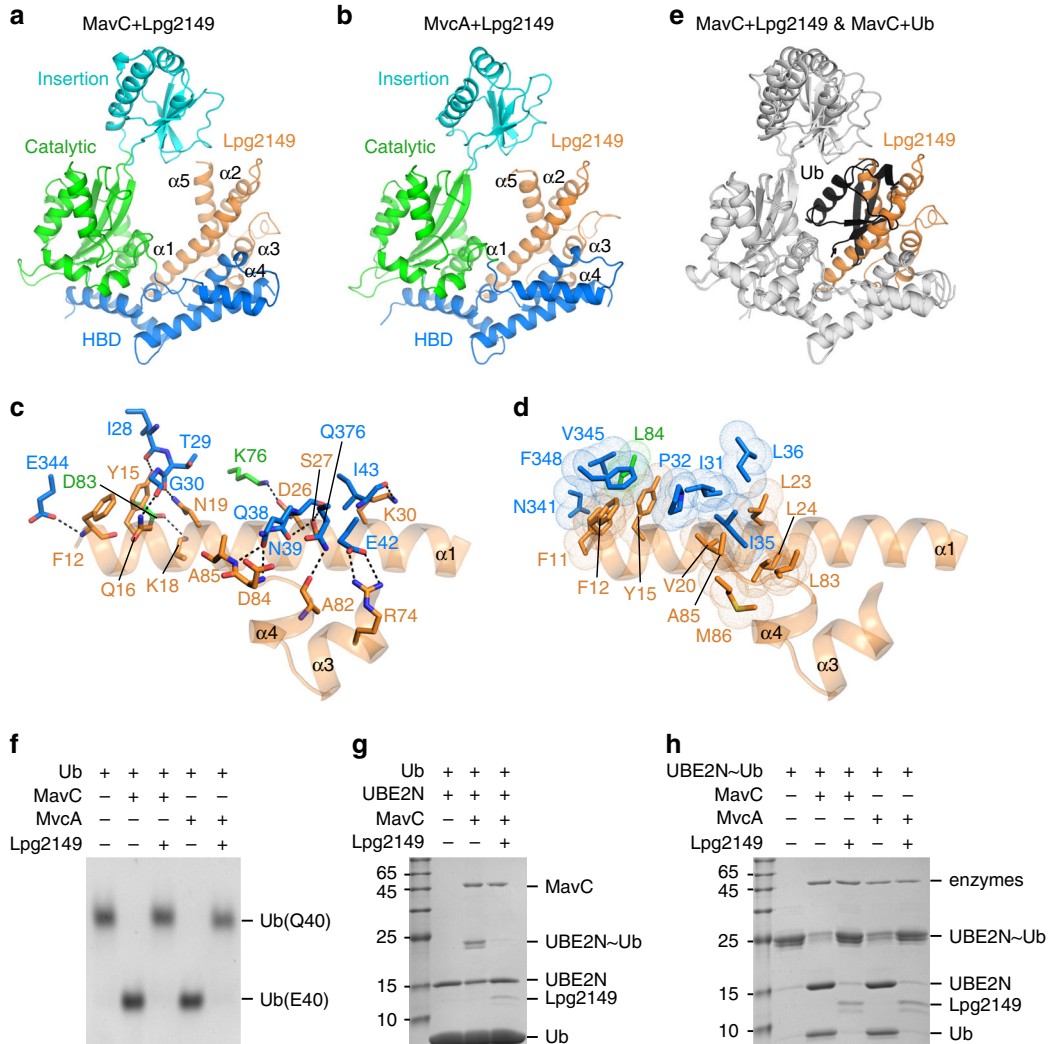

**Fig. 5 Structures of Lpg2149 in complex with MavC and MvcA. a**, **b** Overall structures of the Lpg2149+MavC (**a**) and Lpg2149+MvcA (**b**) complexes. Lpg2149 was shown in orange. MavC and MvcA has the same color code as in Fig. 1a. **c**, **d** Detailed hydrogen bonding (**c**) and hydrophobic (**d**) interactions between Lpg2149 and MavC, with the same color code as in **a**. **e** Structural superimposition between MavC+Lpg2149 and MavC+Ub complexes by using MavC (gray) as the reference point. Lpg2149 and Ub were shown in orange and black, respectively. **f** Lpg2149 can inhibit the Ub deamidation activity of both MavC and MvcA. **g** Lpg2149 can inhibit the UBE2N ubiquitination activity of MavC. **h** Lpg2149 can inhibit the UBE2N~Ub deubiquitination activity of both MavC and MvcA. Source data are provided as a Source Data file. Experiments in **f**–**h** were repeated independently three times with similar results.

(by MavC/MvcA). Indeed, the in vitro biochemical assays confirm this general inhibitory role of Lpg2149 against three different enzymatic activities (Fig. 5f–h).

**Dimer-to-monomer transition of Lpg2149**. Distinct from the monomeric architecture in MavC- and MvcA-bound structures (Fig. 5a, b), Lpg2149 adopts a compact homodimer in its apo form[12]. Here, the extended C-terminal α-helix of one monomer binds within a groove of another monomer, thus forming a domain-swapped dimer assembly (Fig. 6a). Upon binding to the enzymes, this C-terminal α-helix folds back and occupies the groove of the very same Lpg2149 molecule (Fig. 6a). Structural superimpositions between Lpg2149 in its apo and enzyme-bound states show that MavC and MvcA will form heavy steric clashes with the dimeric Lpg2149 (Supplementary Fig. 5d, e), thus providing a structural explanation of this dimer-to-monomer conformational transition.

**Different features of MavC/MvcA binding to Lpg2149**. To see whether Lpg2149 induces conformational changes on MavC and MvcA, we compared the structures of MavC and MvcA in their apo and Lpg2149-bound forms (Fig. 6b). The Catalytic–HBD dual domain of MavC adopts a similar conformation with or without Lpg2149 (Fig. 6b). However, MvcA undergoes a considerable conformational change by enlarging the relative angle between Catalytic and HBD domains to accommodate Lpg2149 (Fig. 6b), indicating an induced-fit model for MvcA. Given that MavC has the pre-formed Lpg2149 binding surface, we anticipated that MavC may have a stronger binding affinity to Lpg2149 than MvcA. Isothermal titration calorimetry (ITC) experiments confirm that the binding of Lpg2149–MavC (~0.57 nM) is ~50-fold stronger than that of Lpg2149–MvcA (~28.08 nM) (Fig. 6c and Supplementary Fig. 6a).

To evaluate the importance of Lpg2149–enzyme interactions (Fig. 5c, d and Supplementary Fig. 5a, b), we generated multiple Lpg2149 mutants covering the inter-molecular surfaces, and

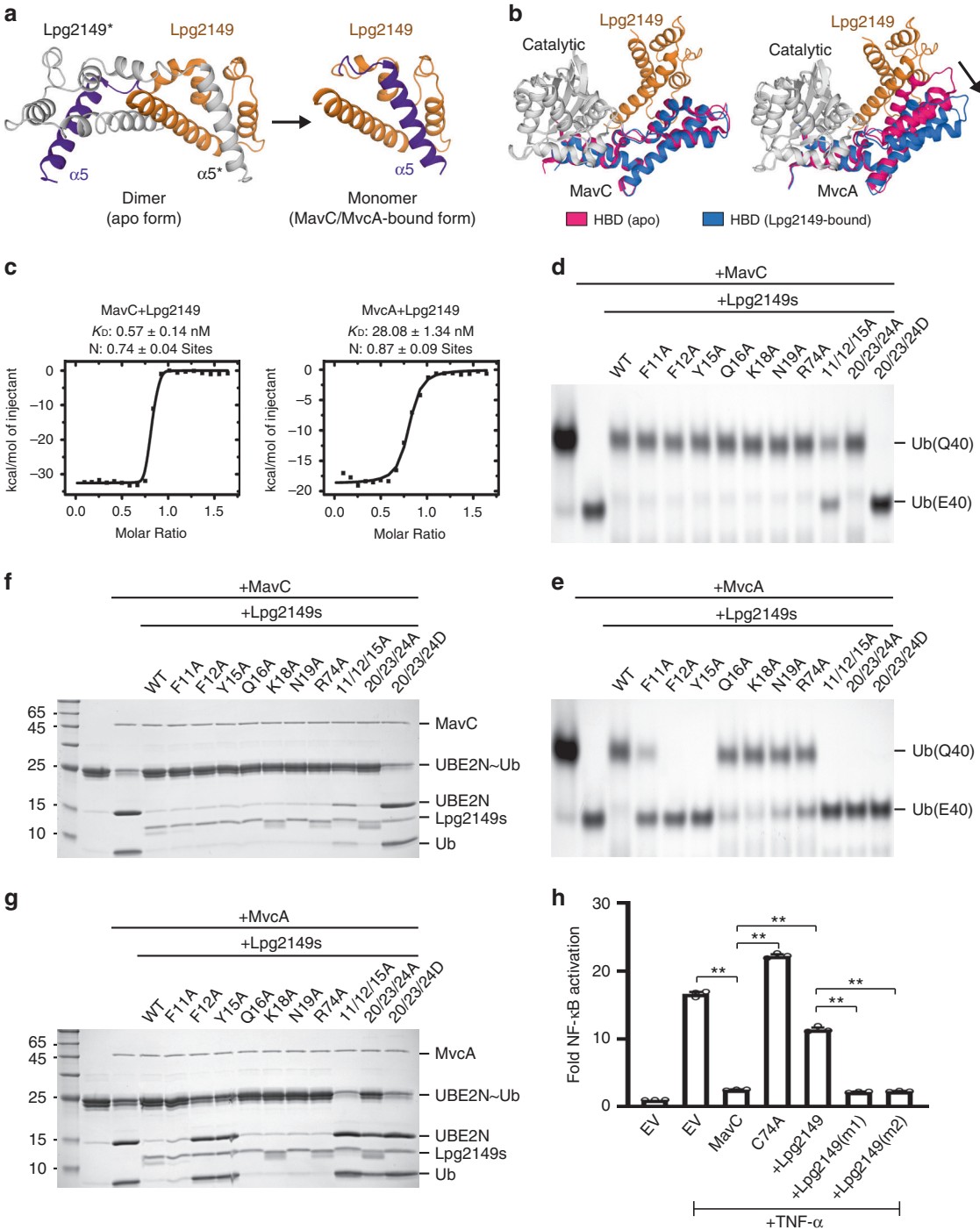

**Fig. 6 Unique inhibiting strategies of Lpg2149 against MavC and MvcA. a** The dimer-to-monomer transition of Lpg2149 upon binding to MavC/MvcA. **b** Conformational changes of MavC (left) and MvcA (right) in their apo and Lpg2149-bound states. The Catalytic domain is the reference point of structural superimpositions for both proteins. **c** Isothermal titration calorimetry (ITC) assays to test binding of Lpg2149 to MavC (left) and MvcA (right). $K_{D}$s and $N$s are mean ± SEM ($n = 2$). **d, e** Inhibition activities of various Lpg2149 mutants for **d** MavC- and **e** MvcA-mediated Ub deamidation. **f, g** Inhibition activities of various Lpg2149 mutants for **f** MavC- and **g** MvcA-mediated UBE2N~Ub deubiquitination. **h** Lpg2149 antagonized MavC's inhibitory effect on NF-κB activation in cells, and the mutations of Lpg2149 (m1: F11D/F12D/Y15D; m2: V20D/L23D/L24D) decreased the antagonism. Plasmids encoding Lpg2149 (or its mutants) and MavC were co-transfected with NF-κB luciferase reporter system plasmids. Cells were mock-treated or treated with TNF-α. The luciferase activities were measured for the activation of NF-κB reporter (see details in Methods). Data presented are means ± SEM of three independent experiments. **$p < 0.01$, unpaired and two-tailed $t$ test. Source data are provided as a Source Data file. Experiments in **c–h** were repeated independently three times with similar results.

analyzed their inhibition activities for both enzymes. Since Lpg2149 binds more strongly to MavC (Fig. 6c and Supplementary Fig. 6a), the mutations of Lpg2149 may have less effects on the inhibition reduction against MavC than that of MvcA. As

expected, Ub deamidation experiments show that most Lpg2149 single-point mutants can still fully inhibit MavC, and only a few triple mutants reduce the inhibition efficiency (Fig. 6d). On the other hand, the same set of Lpg2149 mutants show considerable

reductions for the inhibition of MvcA-mediated Ub deamidation (Fig. 6e). In line with the Ub deamidation results, most Lpg2149 mutants also maintain inhibitory activity against the UBE2N~Ub deubiquitination catalyzed by MavC (Fig. 6f), but show more significant reductions for inhibiting MvcA-mediated reactions (Fig. 6g). Taken together, these data indicate that although Lpg2149 employs a similar interacting pattern to inhibit MavC and MvcA, the details of the inhibiting strategies against these two enzymes are different.

To validate the potential effects of Lpg2149 mutations for the inhibition of MavC-mediated UBE2N ubiquitination, we conducted both in vitro enzymatic analysis and cellular NF-κB activation experiments. Similar to the deamidation (Fig. 6d) and deubiquitination (Fig. 6f) results, the UBE2N ubiquitination activity of MavC can still be strongly inhibited by multiple Lpg2149 mutants except the two triple mutants, F11A/F12A/Y15A and V20D/L23D/L24D (Supplementary Fig. 6b). Further SEC experiments show that the V20D/L23D/L24D mutant cannot form a stable complex with MavC like WT Lpg2149 (Supplementary Fig. 6c). In line with the in vitro results, NF-κB luciferase reporter experiments show that although WT Lpg2149 can rescue the MavC-mediated inhibition of NF-κB signaling, the two triple mutants cannot (Fig. 6h). Immunoblotting experiments confirm that WT Lpg2149 can efficiently inhibit the MavC-mediated ubiquitination of UBE2N, the two triple mutants cannot (Supplementary Fig. 6d).

## Discussion

Thus far, MavC and SidE represent the only two enzyme families that catalyze ubiquitination independent of the well-known E1–E2–E3 cascade. Although both enzymes are encoded by the same bacterial pathogen *L. pneumophila*, their catalysis and regulation mechanisms are distinct from each other. Given that SidE enzymes have been extensively studied, a more detailed research of MavC, shown in the current work, will be essential to understand the diversity of non-canonical ubiquitination systems.

An unexpected finding of the current work is that MavC can deubiquitinate the conjugated UBE2N~Ub generated by MavC itself (Fig. 4a and Supplementary Fig. 4a), in addition to its deamidation and ubiquitination activities. Even more strikingly, MavC catalyzes these three different reactions by employing the same active pocket. It should be notated that the formation and cleavage of a certain type of ubiquitination are always catalyzed by different enzymes in previously reported systems, no matter the conventional E1–E2–E3 systems or the non-canonical SidE system. Although DupA, a recently identified DUB of SidE systems, can switch its activity toward SidE-type Ub ligase by introducing certain mutations, the WT DupA only possess the DUB activity[21]. To our knowledge, MavC represents a unique example of a WT enzyme that possesses the activities of both formation and cleavage of the same type of ubiquitination modification. Although MavC can catalyze two opposite reactions, the overall functional output of MavC in host cells is the ubiquitination on UBE2N (Figs. 3l and 6h), which is likely due to the different efficiencies of the two opposite activities. Interestingly, MvcA, a MavC homolog that is unable to catalyze UBE2N ubiquitination, can remove the MavC-mediated ubiquitination with a much higher efficiency than MavC (Fig. 4h and Supplementary Fig. 4d, e). As a consequence, MvcA acts as the major DUB to remove MavC-mediated UBE2N ubiquitination. In line with our results, a very recent work focusing on MvcA also confirms the DUB activity of MvcA and reports the differential expression patterns of MavC and MvcA during *L. pneumophila* infection for the temporal regulation of the NF-κB activity[28].

One unanswered question regarding MavC and MvcA is why these two structurally similar enzymes have divergent activities. Given that the Catalytic and HBD domains of MavC and MvcA share a similar fold with the conventional Cif-type Ub/NEDD8 deamidases[12,23], both enzymes possess the intrinsic Ub deamidation activity. To gain the additional UBE2N ubiquitination activity, MavC evolved two specific UBE2N interacting surfaces, including an UBE2N-binding groove in the Catalytic domain and a unique Insertion domain that contributes to the tight binding with UBE2N (Fig. 2a). Conversely, MvcA contains the conserved UBE2N-binding surface in the Catalytic domain, but its Insertion domain does not binds UBE2N as strongly as that of MavC (Supplementary Fig. 2f and Fig. 2i). Thus, only MavC, but not MvcA, can form a stable complex with the free-form UBE2N (Fig. 2h), which hinders the ubiquitination activity of MvcA. However, if a conjugated UBE2N~Ub is formed, both the Ub- and UBE2N-mediated interactions will contribute to the overall binding between UBE2N~Ub and MvcA, which will render MvcA sensitive to UBE2N~Ub (Supplementary Fig. 4c). Thus, both MavC and MvcA possess the deubiquitination activity against UBE2N~Ub. Structural comparison of MavC+UBE2N~Ub and MvcA+UBE2N~Ub[28] complexes shows that the Catalytic–HBD regions and the Ub molecules can be superimposed very well, while the Insertion domains and the UBE2N molecules have a considerable rotation between these two structures (Supplementary Fig. 7a). In line with the sequence differences (Supplementary Fig. 2f), the secondary structure elements on the UBE2N-interacting surfaces of the Insertion domains are also not conserved (Supplementary Fig. 7a). Interestingly, although the Insertion domain of MvcA cannot form a stable complex with the free-form UBE2N, it still contributes to the interactions between MvcA and the conjugated UBE2N~Ub[28]. The precise activity control of MavC and MvcA highlights both the power of evolution and the sophisticated strategies employed by the pathogen to achieve successful infections.

UBE2N works together with its partner proteins such as Mms2 to form the canonical K63-linked Ub chains, which regulate multiple cellular processes including the NF-κB pathway[24]. MavC-mediated ubiquitination has been shown to inhibit the canonical E2 activity of UBE2N[9], however, the structural basis of this effect has not been fully discussed. Given that K92 (MavC-mediated ubiquitination site) and C87 (catalytic residue for canonical E2 activity) of UBE2N are in close proximity, the non-canonical ubiquitination at K92 may sterically hinder the canonical E2 reaction. Indeed, structural superimposition of the conjugated UBE2N~Ub in the current structure and the UBE2N–Mms2–Ub complex[26] shows that the Ub molecule in UBE2N~Ub will form severe steric clashes with the donor Ub in the UBE2N–Mms2–Ub complex (Supplementary Fig. 7b). Given that UBE2N is involved in other important cellular processes other than regulating NF-κB pathway, further studies will be needed to fully elucidate the cellular functions of MavC.

It is a common strategy that bacterial pathogens encode both enzymes for generating diverse modifications on host substrates and also their negative regulators for controlling the enzymatic activities. For example, the *Legionella* effector SidJ can catalyze glutamylation on the catalytic glutamate in the mART domain of SidE enzymes, thus blocking their non-canonical ubiquitination activity[17–20]. Distinct from SidJ, Lpg2149 directly binds to MavC and MvcA and occupies the Ub binding site. In this way, Lpg2149 can efficiently inhibit the deamidation, ubiquitination, and deubiquitination activities through one simple mechanism. Moreover, while SidJ only inhibits the ubiquitin ligases but not DUBs in the SidE system, Lpg2149 can inhibit both enzymes in the MavC system.

In summary, our work reveals detailed structural and molecular mechanisms of the non-canonical ubiquitination catalyzed by MavC, covering the Ub recognition, substrate recognition, deamidation/ubiquitination/deubiquitination catalyzation, as well as MvcA- and Lpg2149-mediated negative regulation.

## Methods

**Plasmid construction.** The genes encoding MavC (Lpg2147), MvcA (Lpg2148) and Lpg2149, were amplified from the *Legionella pneumophilalem* genome using standard PCR method. The genes encoding ubiquitin and UBE2N were amplified from human genome. All constructs were cloned into a modified pRSFDuet-1 with N-terminal His-sumo tag. Ubiquitin was also constructed into pETDuet-1 vector with N-terminal His tag for the preparation of the conjugated UBE2N~Ub. For cellular assays, MavC was cloned into pCMV-Myc vector (Clontech) while Lpg2149 was designed into pCMV-HA-Flag vector[29]. All point mutants and truncations were prepared from the native construct using standard PCR method. All the constructs were subsequently confirmed by sequencing. Sequences of all the primers used in this study are listed in Supplementary Table 1.

**Protein expression and purification.** The recombination proteins were over-expressed in BL21 (DE3) strain in Lysogeny Broth medium. The cells were grown at 37 °C until $OD_{600}$ reaching approx. 0.8 and then added with 0.5 mM isopropyl β-D-1-thiogalactopyranoside at 20 °C for overnight induction. Cell pellets were resuspended in lysis buffer containing 50 mM Tris–HCl, pH 7.5, 500 mM NaCl, 20 mM imidazole, 5% glycerol, and then lysed by French Press. Lysates were cleared by centrifugation and the supernatant was loaded onto Ni column. The protein samples were washed with the elution buffer containing 50 mM Tris–HCl, pH 7.5, 500 mM NaCl, 500 mM imidazole, 5% glycerol.

For His-sumo tagged protein except ubiquitin, the Ni column fraction was treated with Ulp1 protease and then reloaded onto Ni column to remove the His-sumo tag and Ulp1. The flow through was fractioned, concentrated and changed to Q column buffer (25 mM Tris–HCl, pH 8.5, 50 mM NaCl). The sample was loaded onto Hitrap Q column and eluted with a linear gradient of NaCl (50 mM to 1 M). The Q column fraction was then purified by gel filtration with the storage buffer (20 mM Tris–HCl, pH 7.5, 100 mM NaCl, 1 mM DTT). The final sample was concentrated and stored at −80 °C before use.

For His-sumo tagged ubiquitin, the Ni column flow through after removing His-sumo tag and Ulp1 was directly purified by gel filtration with a buffer containing 20 mM Tris–HCl, pH 7.5, 10 mM NaCl, 1 mM DTT. For His tagged ubiquitin, the Ni column fraction was immediately loaded to gel filtration chromatography using the same buffer as His-sumo tagged ubiquitin.

All the truncated constructs and mutants were purified following the same protocol used for the preparation of the wild-type proteins.

For producing the conjugated UBE2N~Ub, 5.8 μM UBE2N (or K94A mutant) and 55 μM His-tagged ubiquitin were mixed with 0.45 μM wild-type MavC in a final volume of 200 mL. After incubation at 37 °C for 5 min, the mixture was loaded onto Ni column and the fraction was purified with Q column and gel filtration. The final sample was stored in the buffer containing 20 mM Tris–HCl, pH 7.5, 100 mM NaCl, 1 mM DTT.

**Crystallization.** For crystallization of MavC in complex with UBE2N~Ub, MavC (residues 1–385, C74A) was mixed with the purified UBE2N (K94A)~Ub (1:1.2) at a final concentration of 10 mg mL$^{-1}$. The mixture was incubated at 20 °C for 30 min. The crystals were grown by hanging drop vapor diffusion method at 20 °C, from drops containing 1 μL of the sample and 1 μL of the reservoir solution containing 0.1 M sodium malonate, pH 7.0, 12% PEG3350.

In order to get the crystals of MavC+Lpg2149 complex, Lpg2149 (aa 9–111) was mixed with MavC (aa 1–385) at a molar ratio of 1.5:1. After incubation at 20 °C for 30 min, 1 μL of the sample solution was mixed with 1 μL of the reservoir solution (0.1 M sodium acetate, pH 4.0, 10% PEG4000).

For crystallization of MvcA+Lpg2149 complex, Lpg2149 (aa 9–119) was mixed with MvcA (aa 1–400) at 1.5:1 molar ratio and incubated at 20 °C for 30 min. The crystals were grown by mixing 1 μL of the sample and 1 μL of the reservoir solution (2% Tacsimate, pH 4.0, 0.1 M sodium acetate, pH 4.6, 16% PEG3350).

**Structure determination.** All the diffraction data sets were collected at BL18U1 beamline at the Shanghai Synchrotron Radiation Facility (SSRF). Data were indexed, integrated, and scaled with the HKL program suite[30]. All the structures were solved by molecular replacement using the program PHENIX[31]. To generate the searching models of MavC, the apo MavC structure (PDB: 5TSC) was divided into two parts, the Insertion domain and the Catalytic–HBD dual domain. To generate the searching models of MvcA, the apo MvcA structure (PDB: 5SUJ) was divided into three parts, the Insertion domain, the HBD, and the Catalytic domain. The searching model of Lpg2149 was generated by removing the residues 90–119 from the Lpg2149 apo structure (PDB: 5DPO). The searching models of ubiquitin and UBE2N were derived from the structures of PDB: 5ZQ3 and PDB: 5EYA, respectively. Model building and structural refinement for all the structures were carried out using the programs COOT[32] and PHENIX[31], respectively. The statistics

of the data collection and refinement are shown in Table 1. Structure figures were prepared using Pymol[33].

**In vitro enzymatic assays.** All the in vitro enzymatic reactions were carried out at 37 °C under specific conditions mentioned below.

For the ubiquitination of UBE2N catalyzed by MavC, the total volume of the reaction system is 10 μL, which contains 55 μM ubiquitin, 5.8 μM UBE2N, and 0.45 μM MavC. The reaction buffer contains 20 mM Tris–HCl, pH 7.5, 10 mM NaCl, 5 mM Mn$^{2+}$, 1 mM DTT. For the time-dependent ubiquitination reaction of MavC (Supplementary Fig. 4a), the reaction mixture was added with 2 μL of 6× SDS-PAGE loading buffer and heated at 94 °C for 5 min to stop the reaction at each indicating time point. 10 μL of each sample was subjected to SDS-PAGE and stained with Coomassie dye. For the mutants of MavC (Figs. 2f, g and 3g–i), UBE2N (Supplementary Fig. 2e), or ubiquitin (Fig. 3j), the 10 μL reaction system contains the same amount of mutants as the wild-type proteins. The reactions were stopped after 3 min incubation by adding 2 μL of 6× SDS-PAGE loading buffer and boiling at 94 °C for 5 min. 10 μL of each sample was subjected to SDS-PAGE and stained with Coomassie dye.

For the deamidation activity assay, the 10 μL reaction system contains 55 μM ubiquitin and 0.45 μM MavC (or 0.45 μM MvcA) in the buffer containing 20 mM Hepes, pH 7.5, 10 mM NaCl, 1 mM DTT. After a 2 h incubation, the mixture was added with 2 μL of 6× native PAGE loading buffer, and immediately subjected to the 8% native PAGE, which was followed by staining with Coomassie dye. The mutants of ubiquitin (Fig. 3b and Supplementary Fig. 3d), MavC (Fig. 3k), and MvcA (Supplementary Fig. 3c) were analyzed under the same reaction condition as the wild-type proteins.

For the deubiquitination activity assay, the 10 μL reaction system contains 9.5 μM conjugated UBE2N~Ub and 0.45 μM MavC (or 0.45 μM MvcA) in the buffer containing 20 mM Hepes, pH 7.5, 10 mM NaCl, 1 mM DTT. For the time-dependent deubiquitination reactions of MavC (Supplementary Fig. 4d) or MvcA (Supplementary Fig. 4e), the reaction mixture was added with 2 μL of 6× SDS-PAGE loading buffer and heated at 94 °C for 5 min to stop the reaction at each indicating time point. 10 μL of each sample was subjected to SDS-PAGE and stained with Coomassie dye. For the analysis of mutants, the reaction time is 20 min for MavC (Fig. 4a, c, d) and 20 s for MvcA (Fig. 4e, f), and then analyzed with the same protocol as the wild-type proteins.

**In vitro assays of Lpg2149-mediated inhibition.** For inhibition to ubiquitination activity of MavC, Lpg2149 or its mutants were firstly incubated with MavC (Fig. 5g and Supplementary Fig. 6b) at a molar ratio of 1.5:1 at 20 °C for 30 min in the ubiquitination reaction buffer (20 mM Tris–HCl, pH 7.5, 10 mM NaCl, 5 mM Mn$^{2+}$, 1 mM DTT), and then mixed with UBE2N and ubiquitin. Totally, the final 10 μL reaction system contains 55 μM ubiquitin, 5.8 μM UBE2N, 0.45 μM MavC, and 0.675 μM Lpg2149 or its mutants. After 3 min, the reaction was stopped by adding 2 μL of 6× SDS-PAGE loading buffer and boiling at 94 °C for 5 min. 10 μL of each sample was subjected to SDS-PAGE and stained with Coomassie dye.

For inhibition to deamidation activity of MavC and MvcA, Lpg2149 or its mutants were firstly incubated with MavC (Figs. 5f and 6d) or MvcA (Figs. 5f and 6e) at a molar ratio of 1.5:1 at 20 °C for 30 min in the deamidation reaction buffer (20 mM Hepes, pH 7.5, 10 mM NaCl, 1 mM DTT), and then mixed with ubiquitin. In total, the 10 μL MavC reaction system contains 55 μM ubiquitin, 0.45 μM MavC (or MvcA), and 0.675 μM Lpg2149 or its mutants. After 2 h, the mixture was added with 2 μL of 6× native PAGE loading buffer, and immediately subjected to the 8% native PAGE, which was followed by staining with Coomassie dye.

For inhibition to deubiquitination activity of MavC and MvcA, Lpg2149 or its mutants were firstly incubated with MavC or MvcA at a molar ratio of 3:1 at 20 °C for 30 min in the deubiquitination reaction buffer (20 mM Hepes, pH 7.5, 10 mM NaCl, 1 mM DTT), and then mixed with conjugated UBE2N~Ub. The final 10 μL reaction system contains 9.5 μM ubiquitinated UBE2N, 0.45 μM MavC (or MvcA) and 1.35 μM Lpg2149 or its mutants. After 20 s of reaction for MvcA (Figs. 5h and 6g) or 20 min for MavC (Figs. 5h and 6f), the mixture was added with 2 μL of 6× SDS-PAGE loading buffer and heated at 94 °C for 5 min. 10 μL of each sample was subjected to SDS-PAGE and stained with Coomassie dye.

**Gel filtration chromatography.** Firstly, all the protein samples were changed into the gel filtration buffer (20 mM Tris–HCl, pH 7.5, 100 mM NaCl, 1 mM DTT). The free form UBE2N was mixed with MavC (or its mutants), MvcA, MavC Insertion domain, or MvcA Insertion domain at a molar ratio of 1.5:1. The conjugated UBE2N~Ub was mixed with the MvcA (C83A) mutant at a molar ratio of 1.5:1. Lpg2149 or its mutants were mixed with MavC at a molar ratio of 1.5:1. Each mixture was incubated at 20 °C for 30 min, and loaded onto Superdex 200 increase column (GE Healthcare) equilibrated with the gel filtration buffer. The elution was analyzed with SDS-PAGE and stained by Coomassie dye. Regarding the binding competition experiments of Lpg2149 and conjugated UBE2N~Ub, UBE2N~Ub was first incubated with MavC (C74A) or MvcA (C83A) for 30 min, then followed by the addition of excess Lpg2149. The mixture was loaded onto Superdex 200 increase column (GE Healthcare) equilibrated with the gel filtration buffer and the peak fraction was checked by SDS-PAGE.

**Isothermal titration calorimetry assay**. Lpg2149, MavC, and MvcA were exchanged into the same buffer (20 mM Tris–HCl, pH 7.5, 100 mM NaCl, 1 mM DTT). Binding affinity data were obtained using isothermal titration calorimetry (MicroCal ITC-200) and carried out at 25 °C. MavC or MvcA was thermostated in the sample cell, and Lpg2149 was then injected stepwise over 20 injections with 120 s space apart. The concentrations of MavC, MvcA, and Lpg2149 were determined as 5, 5, and 40 μM, respectively. Data were analyzed by using Origin software (Origin Laboratory).

**Mass spectrometric analysis**. Protein bands corresponding to the ubiquitin molecule after MavC- or MvcA-mediated cleavage (Fig. 4a, e) were manually excised, and performed in-gel digestion with trypsin. Then the sample was transferred for LC–MS/MS analysis.

All nano LC–MS/MS assay were conducted on Q Exactive mass spectrometer (Thermo Scientific) equipped with an Easy n-LC 1000 HPLC system (Thermo Scientific). The digested peptides were separated by a C18 reverse phase column (75 μm × 20 cm, 3 μm). The solvent A contains 0.1% formic acid in water solution and the solvent B consisted of 0.1% formic acid in acetonitrile solution. The gradient was set as 4–8% B, 8 min; 8–22% B, 50 min; 22–32%, B, 12 min; 32–90% B, 1 min; 90% B, 7 min. The flow rate was set at 300 nL min$^{-1}$.

The purified peptides were analyzed on Q Exactive mass spectrometer (Thermo Scientific). The dynamic exclusion time was 40 s. The spray voltage was 2.0 kV and the capillary was heated to 320 °C. With the data-dependent acquisition mode, the MS data were acquired across the mass range of 300–1600 $m/z$ at a high resolution 70,000 ($m/z$ 200). The top 20 precursor ions were selected from each MS scan with isolation width of 2 $m/z$ for fragmentation in the HCD collision cell and the MS/MS spectra were acquired at resolution 17,500 ($m/z$ 200).

The raw data were analyzed by using Thermo Proteome Discoverer (version 1.4.0.228) with Sequest HT search engine for peptide identification against a manually-build database containing wild type ubiquitin, the Q40E mutant, UBE2N and other random proteins. When searching, trypsin was selected as enzyme and two missed cleavages were allowed; the mass tolerance of precursor was set as 10 ppm and the ions tolerance was 20 mDa. False discovery rate analysis was performed with Percolator and was set <1% for protein identification. The peptides confidence was set high for peptides filter.

**NF-κB luciferase assay**. NF-κB controlled *firefly* luciferase expression plasmid pGL3-NFκB-luc[34] was used as the reporter system, and the plasmid pRL-TK (Promega) expressing *renilla* luciferase was used as an internal control reporter for normalization. HEK293T cells (ATCC CRL-11268) were seeded on 12-well plates at $3 \times 10^5$/well. For mutagenesis assay of MavC, 200 ng empty vector (pCMV-myc), wild-type MavC or MavC mutant vector were respectively co-transfected with 100 ng pGL3-NFκB-luc and 10 ng pRL-TK. For Lpg2149 inhibition assay, 200 ng MavC vector and 800 ng Lpg2149 (or its mutants) vector were co-transfected together with 100 ng NF-κB plasmid and 10 ng pRL-TK vector. At 24 h post-transfection, cells were mock-treated or treated with 20 ng mL$^{-1}$ TNF-α. 5 h later, cells were lysed for luciferase assay with a Dual-Luciferase Reporter Assay System (Promega) according to the manufacturer's instructions.

**Western blotting**. For K63-linked polyubiquitination assays, HA-ubiquitin-K63 (all other Lys residues were mutated to Arg except K63) and myc-MavC or its mutants were cotransfected into cells. For all western blotting assays, HEK293T cells were collected at 28 h posttransfection and lysed with RIPA buffer (150 mM NaCl, 50 mM Tris–HCl, pH 7.5, 1% NP40, and 1% SDS). Cell lysates were resolved by SDS-PAGE gels and transferred to PVDF membranes for immunoblotting analysis with specific antibodies: mouse monoclonal Flag-specific antibody M2 (Sigma-Aldrich, Cat# F3165), 1:10,000; mouse monoclonal myc-specific antibody 9E10 (Santa Cruz Biotechnology, Cat# SC-40), 1:5000; mouse monoclonal β-actin-specific antibody (Sigma-Aldrich, Cat# a5316), 1:10,000; HA tag rabbit polyclonal antibody (Proteintech, Cat# 51064-2-AP), 1:5000; UBE2N (D2A1) rabbit mAb (Cell Signaling Technology, Cat# 6999S), 1:3000.

**Quantification and statistical analysis**. The band intensities in Supplementary Fig. 4d, e were measured with ImageJ software, and then the data were processed by Log function and analyzed with GraphPad Prism. The NF-κB luciferase signal (Figs. 3l and 6h) were normalized to *renilla* luciferase signal, figured out the growth ratio after stimulation, and analyzed with GraphPad Prism. The error bars shown in these figures represent mean ± SEM. Statistical significance was determined by unpaired and two-tailed $t$ test in Excel: *$p < 0.05$, **$p < 0.01$, and ***$p < 0.001$.

**Reporting summary**. Further information on research design is available in the Nature Research Reporting Summary linked to this article.

## Data availability
The accession numbers for the coordinates and structure factors reported in this paper are PDB 7BXG (MavC+UBE2N~Ub), PDB 7BXH (MavC+Lpg2149), and PDB 7BXF (MvcA +Lpg2149). The source data underlying Figs. 2f–i, 3g–l, 4a, c–f, 5f–h, and 6d–h and Supplementary Figs. 2d, e, 3b–f, 4a–e, 5c, and 6b–d are provided as a Source Data file. All data and materials are also available from the corresponding author upon request.

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

## Acknowledgements

The authors thank the staff of BL17U1/BL18U1/BL19U1 beamlines at National Center for Protein Sciences Shanghai and Shanghai Synchrotron Radiation Facility, Shanghai, People's Republic of China, for assistance during data collection. The authors thank Dr. Ang Gao for the critical reading and helpful discussion. The authors thank Dr. Fuquan Yang, Dr. Jifeng Wang, and Dr. Yuanyuan Chen for their support with the mass spectrometry and ITC assays. This work was supported by grants from the National Natural Science Foundation of China (91753133, 31670903, 31922037, and 31700687), the National Key R&D Program of China (2018YFA0508000 and 2018YFA0507203), the Chinese Academy of Sciences Pilot Strategic Science and Technology Project B (XDB37030203), the Thousand Young Talents Program, and the Youth Innovation Promotion Association (2019097).

## Author contributions

Y.W. and Q.Z. conducted crystallographic and biochemical experiments with the help of P.L. and S.L. X.W. performed the cellular experiments with the help of Y.W. and Q.Z. G.G. and P.G. directed the research. Y.W. and P.G. wrote the manuscript with the help of other authors.

## Competing interests

The authors declare no competing interests.
