## [Peer Review File · Nature Communications]

Reviewers' comments:

Reviewer #1 (Remarks to the Author):

The paper by Wang et al., describes the structure and function of two legionella effector proteins MavC and MvcA. The presented crystal structures and biochemistry quite comprehensively explain opposing activities of MavC and MvcA and inhibitory role of Lpg2149. This paper together with Gan et al., 2019 provides structural basis for another atypical ubiquitination adopted by Legionella and will be of general interest to readers in the ubiquitin field and especially to researchers investigating legionella effectors.

following are my concerns with the paper

1. In figure 3, authors present the lack of ubiquitination activity of several of structure based point mutants of MavC. But in NF- κ B luciferase reporter assay presented in Figure 3I, these MavC mutants still were very effective in inhibiting Nf- κ B. Authors should explain this discrepancy. Perhaps the effect of structure based point mutants can be better described in the context of legionella infection?
2. UBE2N is involved in other important cellular processes, does MavC also affect those or is its inhibition specific to NF- κ B?
3. In addition to the inhibition assays performed in Figure 5, authors should perform binding competition assays to see if Lpg2149 can replace UBE2N-Ub bound to MavC/MvcA.
4. Although, the authors nicely explained why MvcA cannot be a ubiquitin ligase due to its lack of binding to UBE2N alone, it is still not clear why MvcA has more DUB activity than MavC. Can this be discussed/explained in discussion section (line 337)?
5. in line 222, authors point to MavC having DUB activity as a new phenomenon. In fact, such dual and opposing activity has been recently reported for another legionella effector DupA (Shin et al., 2019 Mol Cell) and should be mentioned.

Reviewer #2 (Remarks to the Author):

In the submitted paper “Insights into catalysis and regulation of deamidase-mediated non-canonical ubiquitination and deubiquitination”, the authors investigate a system of bacterial effector proteins that mediate deamidation, ubiquitination and deubiquitination of specific eukaryotic target proteins.

Specifically, the authors show that the *Legionella* effector protein MavC can ubiquitinate the eukaryotic cellular substrate Ube2N, in the absence of canonical ubiquitin pathway related proteins (ie eukaryotic E1, E2 and E3s). This finding was also described recently by Valteau et.al (2018) Cell Reports and Gan et al (2019) Nature Microbiology.

The manuscript describes the crystal structure of MavC bound to Ube2N and Ub, essentially the ubiquitination reaction product comprising an isopeptide linkage between Gln40 of ubiquitin and Lys92 of Ube2N. This protein complex is highly complementary to the related Mvca-Ube2N-Ub complex (PDB: 6JKY, Gan et al, 2019, EMBO J.). The notable difference is that Mvca only performs the reverse of a MavC ubiquitination reaction on Ube2N (whereas MavC can catalyze both the forward and back reaction).

Structure guided-mutants were designed and tested in ubiquitination assays to validate the functional relevance of the contacts between MavC, Ube2N and Ub. In addition, native-PAGE and mass spectrometry analysis confirmed that deamidation of Gln40-Ub is mediated by MavC in the absence of Ube2N. The results are consistent with their structural hypothesis that mutations that disrupt MavC interaction with Ube2N or Ub lead to compromised ubiquitination activity.

An insertion region within MavC, that is divergent with the analogous region in Mvca, contains several interacting residues with Ube2N. Mutations within this region compromise deubiquitination activity by both MavC and Mvca. Using time-course deubiquitination experiments, the authors show that although MavC and Mvca use the same catalytic mechanism to deubiquitinate Ube2N, the activity of Mvca is more robust.

The following is the most novel and interesting aspect of the storyline.

The authors next confirmed that Lpg2149 inhibits MavC and Mvca enzymatic activity by binding to the active cleft of these enzymes. The authors were the first to show the basis for this inhibition by

crystallizing Lpg2149 in complex with MavC and separately with MvcA. Although Lpg2149 adopts a dimeric structure in its apo form (PDB: 5DPO shown previously by Valteau et al), Lpg2149 is monomeric when bound to MavC or MvcA.

Finally, the authors used a luciferase reporter assay to assess whether MavC and Lpg2149 can influence NF- κ B signalling in cells (a function reported previously by other groups). Expression of wild-type MavC in cells inhibited NF- κ B activation (as expected). When key contact residues in MavC were mutated to impair the MavC-Ube2N interaction, MavC was also no longer able to inhibit the activation of the NF- κ B pathway in cells. When Lpg2149 was mutated to impair the Lpg2149-MavC interaction, Lpg2149 was no longer able to modulate the inhibition of NfKb signalling by MavC in cells (comparable to cells lacking Lpg2149). The results relating to Lpg2149 are novel (not reported elsewhere).

Overall the work described in the submitted manuscript is interesting, the structural complexes are novel (especially relating to Lpg2149) and highly complementary to the body of work described recently by other groups. As such, I recommend its publication in Nature Comm, after the authors address the following points.

1. In validating the structural complexes, the authors make extensive use of point mutants for functional assays (either in vitro or the cell based NfKB readouts). No data is presented that demonstrates the mutations have an effect on the binding of MavC to Ube2N, Ub or Lpg2149. This reviewer recommends the authors perform binding assays to confirm the loss of activity observed is consistent with compromised binding for some select mutants.

- 2 In their abstract, the authors state that MavC ubiquitination of Ube2N abolishes the ability of Ube2N to form K63-linked Ub chains, therefore dampening NF- κ B signaling. Figure 3i shows that wt-MavC dampens NF- κ B signaling in cells. However, the authors have not actually shown that the MavC ubiquitinated form of Ube2N occurs in cells and is reduced in the presence of MavC-C74A where NfKb signaling is not activated.

As such, the possibility remains that binding of MavC (wt and C74A) to Ube2N may be responsible for the effects on NfKb signalling (without need for catalysis). The author also does not show that the levels of K63-Ub chains are affected in cells. Is K63 chain formation impacted by the expression of MavC vs MavC-C74A?

3. Furthermore, is the MavC ubiquitinated form of Ube2N decreased in cells upon co-expression of Lpg2149?

4. In Fig. 3h, the authors claim that there is a reduction of Ube2N ubiquitination when residues mediating the interaction between the HBD domain of MavC and Ub are mutated. However, the results are very weak. Can the authors quantitate the changes in levels of Ube2N~Ub in these experiments? How reproducible are the effects? Perhaps double or triple mutants would give a more compelling effect at this interface.

5. Not all of the tested mutants shown in Fig. 3h are visible in the structural panels above. Can the authors add these?

6. The authors should include a superimposed figure comparing the structures of Mvca-Ube2N-Ub (by Gan et al) with MavC-Ube2N-Ub (in this paper). This analysis will provide further insights into the catalytic differences between MavC and Mvca.

We thank the reviewers for their valuable comments. In the revised manuscript, we have performed more experiments and carefully addressed the reviewers' concerns. We believe that the revised version, which takes into account suggestions of the reviewers and recommendations of the editor, is well suited for *Nature Communications*. The changes in the manuscript has been highlighted as underlined text. Our detailed responses to reviewers' critiques are presented below (reviewers' comments are in red and our responses are in black).

Reviewers' comments:

Reviewer #1 (Remarks to the Author):

The paper by Wang et al., describes the structure and function of two legionella effector proteins MavC and MvcA. The presented crystal structures and biochemistry quite comprehensively explain opposing activities of MavC and MvcA and inhibitory role of Lpg2149. This paper together with Gan et al., 2019 provides structural basis for another atypical ubiquitination adopted by Legionella and will be of general interest to readers in the ubiquitin field and especially to researchers investigating legionella effectors.

We thank the reviewer for summarizing the key contributions of our work and finding this paper "will be of general interest to the readers" in both ubiquitin and legionella fields.

following are my concerns with the paper

1. In figure 3, authors present the lack of ubiquitination activity of several of structure based point mutants of MavC. But in NF- κ B luciferase reporter assay presented in Figure 3I, these MavC mutants still were very effective in inhibiting Nf- κ B. Authors should explain this discrepancy. Perhaps the effect of structure based point mutants can be better described in the context of legionella infection?

Thank you for pointing out this issue. Unlike the catalytic residue mutant (C74A) that leads to complete inactivation of MavC, the mutants located at inter-molecular interacting surfaces (S73A, R121A, R126A, F188A/Y189A/Y192A) actually cannot completely abolish the activity. Indeed, we still observed the residual activities for these surface mutants in our biochemical assays (Fig. 2 and Fig. 3), and more obvious activities, if the reaction time was extended (below figure). Furthermore, our new immunoblotting experiments confirm that these surface mutants can still ubiquitinate UBE2N in HEK293T cells, although the activities are much lower than wild-type MavC (Supplementary Fig. 3f). Therefore, these surface mutants still maintain partial inhibition abilities in the NF- κ B luciferase reporter assays (Fig. 3I). By following the reviewer's suggestions, we have explained this discrepancy in the revised manuscript (line 196-199).

We fully agree with the reviewer that the effects of all structure based mutants can be better described in the context of legionella infection. However, as a structural biology laboratory, we do not have the resources and license to perform bacterial pathogen infection experiments. In addition, we believe that these experiments extend the scope of the current structural/biochemical paper and may be done in future functional studies. In the revised manuscript (line 200-202), we have acknowledged the importance of legionella infection experiments.

2. UBE2N is involved in other important cellular processes, does MavC also affect those or is its inhibition specific to NF-kB?

Given that NF-kB pathway has been known to be one of the most established targets of *L. pneumophila* effectors, all the previous MavC/MvcA related works (both functional and structural works) only focus on this pathway but not other UBE2N-regulated cellular processes (Valleau et al, *Cell Rep* 2018; Gan et al, *Nat Microbiol* 2019; Gan et al, *EMBO J* 2020). Our current structural and biochemical work is mainly focusing on the catalyzation & regulation mechanisms of MavC/MvcA, and the purpose of NF-kB luciferase reporter assays in this work is only to show the effects of certain mutations. Thus, although it is definitely important to uncover the potential roles of MavC for other cellular processes, this is beyond the scope of the current contribution and should be done in future studies. By following the reviewer's suggestion, we have acknowledged that more work will be needed to fully elucidate the cellular functions of MavC (line 390-392).

3. In addition to the inhibition assays performed in Figure 5, authors should perform binding competition assays to see if Lpg2149 can replace UBE2N-Ub bound to MavC/MvcA.

Thank you for the suggestions. We performed binding competition experiments by using the size exclusion chromatography method (Supplementary Fig. 5c). Specifically, UBE2N~Ub was first incubated with MavC-C74A or MvcA-C83A for 30 min, then followed by the addition of Lpg2149. The mixture was loaded onto the size exclusion column and the peak fraction was checked by SDS-PAGE. As expected, Lpg2149 can compete with UBE2N-Ub for the binding to both MavC and MvcA, albeit with different efficiency. We have added these new data in the revised manuscript (line 278-281).

4. Although, the authors nicely explained why MvcA cannot be a ubiquitin ligase due to its lack of binding to UBE2N alone, it is still not clear why MvcA has more DUB activity than MavC. Can this be discussed/explained in discussion section (line 337)?

Thank you for raising this important question. We anticipated that MvcA may have an intrinsically higher DUB activity than MavC, therefore, MvcA can catalyze the cleavage reaction more efficiently once the conjugated UBE2N~Ub binds to the enzyme sufficiently strongly. In addition, although the formation and cleavage of this non-canonical ubiquitination are catalyzed within the same pocket, the ligase and DUB activities are not necessarily coupled with each other. We have added these points in the revised manuscript and acknowledged that more work will be needed to fully understand this question (line 249-253 and line 257-259).

5. in line 222, authors point to MavC having DUB activity as a new phenomenon. In fact, such dual and opposing activity has been recently reported for another legionella effector DupA (Shin et al., 2019 Mol Cell) and should be mentioned.

Thank you for pointing out this issue. To our knowledge, the WT DupA has only DUB activity, but no ubiquitin ligase activity. A point mutation (E242R) that reduces DupA-ubiquitin binding will switch DupA's activity toward a SidE-type ubiquitin ligase. Thus, despite this mutation-caused new activity of DupA, MavC still represents a unique example of a WT enzyme that possesses the activities of both

formation and cleavage of the same type of ubiquitination modification. By following the reviewer's suggestions, we have added the DupA information in the revised manuscript (line 344-348).

Reviewer #2 (Remarks to the Author):

In the submitted paper “Insights into catalysis and regulation of deamidase-mediated non-canonical ubiquitination and deubiquitination”, the authors investigate a system of bacterial effector proteins that mediate deamidation, ubiquitination and deubiquitination of specific eukaryotic target proteins.

Specifically, the authors show that the Legionella effector protein MavC can ubiquitinate the eukaryotic cellular substrate Ube2N, in the absence of canonical ubiquitin pathway related proteins (ie eukaryotic E1, E2 and E3s). This finding was also described recently by Valteau et.al (2018) Cell Reports and Gan et al (2019) Nature Microbiology.

The manuscript describes the crystal structure of MavC bound to Ube2N and Ub, essentially the ubiquitination reaction product comprising an isopeptide linkage between Gln40 of ubiquitin and Lys92 of Ube2N. This protein complex is highly complementary to the related MvcA-Ube2N-Ub complex (PDB: 6JKY, Gan et al, 2019, EMBO J.). The notable difference is that MvcA only performs the reverse of a MavC ubiquitination reaction on Ube2N (whereas MavC can catalyze both the forward and back reaction).

Structure guided-mutants were designed and tested in ubiquitination assays to validate the functional relevance of the contacts between MavC, Ube2N and Ub. In addition, native-PAGE and mass spectrometry analysis confirmed that deamidation of Gln40-Ub is mediated by MavC in the absence of Ube2N. The results are consistent with their structural hypothesis that mutations that disrupt MavC interaction with Ube2N or Ub lead to compromised ubiquitination activity.

An insertion region within MavC, that is divergent with the analogous region in MvcA, contains several interacting residues with Ube2N. Mutations within this region compromise deubiquitination activity by both MavC and MvcA. Using time-course deubiquitination experiments, the authors show that although MavC and MvcA use the same catalytic mechanism to deubiquitinate Ube2N, the activity of MvcA is more robust.

The following is the most novel and interesting aspect of the storyline.

The authors next confirmed that Lpg2149 inhibits MavC and MvcA enzymatic activity by binding to the active cleft of these enzymes. The authors were the first to show the basis for this inhibition by crystallizing Lpg2149 in complex with MavC and separately with MvcA. Although Lpg2149 adopts a dimeric structure in its apo form (PDB: 5DPO shown previously by Valteau et al), Lpg2149 is monomeric when bound to MavC or MvcA.

Finally, the authors used a luciferase reporter assay to assess whether MavC and Lpg2149 can influence NF-κB signalling in cells (a function reported previously by other groups). Expression of wild-type MavC in cells inhibited NF-κB activation (as expected). When key contact residues in MavC were mutated to impair the MavC-Ube2N interaction, MavC was also no longer able to inhibit the activation of the NF-κB pathway in cells. When Lpg2149 was mutated to impair the Lpg2149-MavC interaction, Lpg2149 was no longer able to modulate the inhibition of NfKb signalling by MavC in cells (comparable to cells lacking Lpg2149). The results relating to Lpg2149 are novel (not reported elsewhere).

Overall the work described in the submitted manuscript is interesting, the structural complexes are novel (especially relating to Lpg2149) and highly complementary to the body of work described recently by other groups. As such, I recommend its publication in Nature Comm, after the authors address the following points.

We thank the reviewer for clearly summarizing the key contributions of our work. We also thank this reviewer for finding our work "interesting and novel" and recommending publication after addressing the critiques. In the revised manuscript, we have performed more experiments and carefully addressed the reviewers' concerns.

1. In validating the structural complexes, the authors make extensive use of point mutants for functional assays (either *in vitro* or the cell based NfKB readouts). No data is presented that demonstrates the mutations have an effect on the binding of MavC to Ube2N, Ub or Lpg2149. This reviewer recommends the authors perform binding assays to confirm the loss of activity observed is consistent with compromised binding for some select mutants.

Thank you for the suggestions. We performed size exclusion chromatography experiments to check the complex formation for some selected mutants on both MavC-UBE2N and MavC-Lpg2149 interfaces. Distinct with the wild-type proteins that can form stable MavC-UBE2N (Fig. 2h) and MavC-Lpg2149 (Supplementary Fig. 6c) complexes, the mutations on both MavC-UBE2N interface (MavC^{Y198A} and MavC^{F188A/Y189A/Y192A}) and MavC-Lpg2149 interface (Lpg2149^{V20D/L23D/L24D}) will disrupt the complex formation under the same conditions (Supplementary Fig. 2d and 6c). We have added these binding results in the revised manuscript (line 132-134 and line 324-325).

On the other hand, even the wild-type Ub and MavC cannot form a stable complex on the size exclusion column, indicating the binding between Ub and MavC is weaker than that of UBE2N-MavC or Lpg2149-MavC. Thus, we tried two other methods to test the binding: pull-down and isothermal titration calorimetry (ITC). Similar to the size exclusion chromatography results, we cannot detect notable binding between wild-type Ub and MavC in these assays (below figure). This is probably also the reason why none of the previous works conducts direct binding between Ub and MavC/MvcA. Due to this technique reason, we still used the functional assays (*in vitro* enzymatic reactions or cell-based NF- κ B readouts) to validate the interactions between Ub and MavC in the revised manuscript.

2. In their abstract, the authors state that MavC ubiquitination of Ube2N abolishes the ability of Ube2N to

form K63-linked Ub chains, therefore dampening NF- κ B signaling. Figure 3i shows that wt-MavC dampens NF- κ B signaling in cells. However, the authors have not actually shown that the MavC ubiquitinated form of Ube2N occurs in cells and is reduced in the presence of MavC-C74A where NfKb signaling is not activated.

As such, the possibility remains that binding of MavC (wt and C74A) to Ube2N may be responsible for the effects on NfKb signalling (without need for catalysis). The author also does not show that the levels of K63-Ub chains are affected in cells. Is K63 chain formation impacted by the expression of MavC vs MavC-C74A?

Thank you for raising these important questions. The previous functional work (Gan et al, *Nat Microbiol* 2019) has already shown that MavC can ubiquitinate UBE2N in cells, while MavC-C74A cannot. To better address the reviewer's concern, we checked the ubiquitination status of UBE2N in HEK293T cells expressing different MavC proteins, including WT MavC, MavC-C74A, MavC-S73A, MavC-R121A, MavC-R126A, or MavC-F188A/Y189A/Y192A. The immunoblotting experiments show that: WT MavC can efficiently ubiquitinate UBE2N; MavC-C74A cannot ubiquitinate UBE2N; the other four mutants on the inter-molecular surfaces can only partially ubiquitinate UBE2N (Supplementary Fig. 3f). We have added these new data in the revised manuscript (line 196-199).

Our size exclusion chromatography experiments show that both WT MavC (Fig. 2h) and MavC-C74A (below figure) can bind to UBE2N. However, only WT MavC, but not MavC-C74A, can inhibit NF- κ B signaling. These results indicate that the MavC-mediated ubiquitination of UBE2N, rather than the binding between MavC and UBE2N, is responsible for the effects on NF- κ B signaling.

We further checked the overall level of cellular K63-Ub chains upon expression of WT MavC or MavC-C74A. The results clearly show that WT MavC, rather than MavC-C74A, causes significant reduction of K63-Ub chain formation (Supplementary Fig. 3e). Please note that other lysine residues except K63 have been mutated to arginine in the transfected HA-ubiquitin-K63 plasmid (Supplementary Fig. 3e). We have added these new data in the revised manuscript (line 192-194).

3. Furthermore, is the MavC ubiquitinated form of Ube2N decreased in cells upon co-expression of Lpg2149?

Thank you for pointing out this issue. Our new immunoblotting experiments confirm that Lpg2149 significantly reduces the amount of UBE2N~Ub (Supplementary Fig. 6d). However, two loss-of-function Lpg2149 mutants (F11D/F12D/Y15D; V20D/L23D/L24D) have no such effects (Supplementary Fig. 6d). We have added these new data in the revised manuscript (line 328-329).

4. In Fig. 3h, the authors claim that there is a reduction of Ube2N ubiquitination when residues mediating the interaction between the HBD domain of MavC and Ub are mutated. However, the results are very weak. Can the authors quantitate the changes in levels of Ube2N~Ub in these experiments? How reproducible are the effects? Perhaps double or triple mutants would give a more compelling effect at this interface.

Thank you for the suggestions. Indeed, some single point mutations on the MavC-Ub interface (e.g. N39A, E42A, E66A, N72A) only slightly reduce the activity, as shown in the original Fig. 3h. Although the reduction effects are reproducible, we do agree with the reviewer that double or triple mutants would be a better approach to validate these interactions. In the revised manuscript, we updated the Fig. 3h panel with two double mutants (N39A/E42A; E66A/N72A). As expected, these double mutants give a more compelling effect than the original single point mutants.

5. Not all of the tested mutants shown in Fig. 3h are visible in the structural panels above. Can the authors add these?

Thank you for pointing out this mistake. We have added the E66-mediated interaction in the revised Fig. 3d.

6. The authors should include a superimposed figure comparing the structures of MvcA-Ube2N-Ub (by Gan et al) with MavC-Ube2N-Ub (in this paper). This analysis will provide further insights into the catalytic differences between MavC and MvcA.

Thank you for the suggestions. Structural comparison of MavC+UBE2N~Ub and MvcA+UBE2N~Ub complexes shows that the Catalytic-HBD regions and the Ub molecules can be superimposed very well, while the Insertion domains and the UBE2N molecules have a considerable rotation between these two structures (Supplementary Fig. 7a). In line with the sequence differences of the Insertion domains (Supplementary Fig. 2f), the secondary structure elements on the UBE2N-interacting surfaces of the Insertion domains are also not conserved (Supplementary Fig. 7a). Interestingly, although the Insertion domain of MvcA cannot form a stable complex with the free-form UBE2N, it still contributes to the interactions between MvcA and the conjugated UBE2N~Ub (Supplementary Fig. 7a) (Gan et al, 2020). In the revised manuscript, we have added these points in the Discussion section (line 370-378).

REVIEWERS' COMMENTS:

Reviewer #1 (Remarks to the Author):

All my concerns have been adequately addressed in the revised version of the manuscript. This work will be of general interest to researchers in the ubiquitin field and to people studying legionella infection biology. I recommend publishing in Nature Communications.

Reviewer #2 (Remarks to the Author):

the authors have addressed all the issues raised in my review.

I recommend publication.